# The Role of αvβ3 Integrin in Cancer Therapy Resistance

**DOI:** 10.3390/biomedicines12061163

**Published:** 2024-05-24

**Authors:** Bianca Cruz Pachane, Heloisa S. Selistre-de-Araujo

**Affiliations:** 1Biochemistry and Molecular Biology Laboratory, Department of Physiological Sciences, Universidade Federal de São Carlos (UFSCar), São Carlos 13565-905, SP, Brazil; bpachane@estudante.ufscar.br; 2Department of Molecular and Comparative Pathobiology, Johns Hopkins University School of Medicine, Baltimore, MD 21205, USA

**Keywords:** tumor resistance, integrins, αvβ3 integrin, tyrosine kinase inhibitors, immune checkpoint inhibitors, radiotherapy resistance

## Abstract

A relevant challenge for the treatment of patients with neoplasia is the development of resistance to chemo-, immune-, and radiotherapies. Although the causes of therapy resistance are poorly understood, evidence suggests it relies on compensatory mechanisms that cells develop to replace specific intracellular signaling that should be inactive after pharmacological inhibition. One such mechanism involves integrins, membrane receptors that connect cells to the extracellular matrix and have a crucial role in cell migration. The blockage of one specific type of integrin is frequently compensated by the overexpression of another integrin dimer, generally supporting cell adhesion and migration. In particular, integrin αvβ3 is a key receptor involved in tumor resistance to treatments with tyrosine kinase inhibitors, immune checkpoint inhibitors, and radiotherapy; however, the specific inhibition of the αvβ3 integrin is not enough to avoid tumor relapse. Here, we review the role of integrin αvβ3 in tumor resistance to therapy and the mechanisms that have been proposed thus far. Despite our focus on the αvβ3 integrin, it is important to note that other integrins have also been implicated in drug resistance and that the collaborative action between these receptors should not be neglected.

## 1. Introduction

The development of therapy resistance in cancer patients is a reoccurring and concerning issue in clinics, arising as a consequence of multifaceted events that involve both external and intrinsic components ranging from therapeutic pressure to tumor burden, heterogeneity, and disease progression. Due to this challenge in treatment, it is common that throughout cancer development patients are exposed to different types of drugs, with or without combination with other compounds and treatment settings, in varying dosages. This aggressive approach can be modulated as the patient response is observed, until a point where the illness has either been cured, has plateaued, or has aggravated [1]. Luckily, the methods for cancer treatment have evolved in such a way that recurrent tumors have become less frequent, yet certain types of cancer remain with a high index of recurrence and mortality. For example, patients in a clinical trial from Iran with recurrent glioblastoma had a median overall survival of 16 months [2]. In the Netherlands, a multicenter cohort study has found that 42% of patients proceeded to develop recurrent colon cancer in the five years following up remission [3].

Recurrent cancer can be developed locally (i.e., in the same tissue), regionally (i.e., in adjacent tissues), or distantly from the primary site, and some types of cancers have a tropism to metastasize towards specific tissues, as is the case in breast-to-bone cancer [4,5]. It is also known that recurrent tumors may not present characteristics similar to the original mass; therefore, their treatment needs to be adjusted to target the new neoplastic microenvironment [6]. Hence, understanding how tumor progression occurs and determining different approaches to mitigate therapy resistance are essential topics for discussion.

In this review, we will focus on the role of integrins, adhesion molecules that have long been investigated as targets and cancer biomarkers [7]. In particular, we will focus on integrin αvβ3 and how they regulate specific cell behaviors that modulate tumor development, progression, and resistance to therapy.

## 2. Integrins in Tumor Progression

Integrins are surface adhesion proteins formed by a large globular extracellular segment, a single transmembrane loop, and a smaller cytoplasmic tail, which are combined in heterodimers that are formed by a set of 18 α subunits and 8 β subunits. Each dimer recognizes and binds, with distinct affinity, to motifs found in extracellular matrix (ECM) components such as vitronectin, collagen, and fibronectin [8] (Figure 1).

Integrins can be activated upon ligand binding in either the intracellular tail, which elicits a modification of the extracellular environment, or the extracellular portion. In this situation, a ligand triggers a cascade of intracellular pathways that lead to growth factor binding of tyrosine kinase cognate receptors. Therefore, although integrins have no intrinsic tyrosine kinase activity, their activation enables their intracellular domain to act as an adaptor to several accessory proteins such as kindling, talin, and the focal adhesion kinase (FAK). Consequently, tyrosine kinase receptor activation supports the integrin adhesome and helps propagate extracellular signals [9,10,11]. The downstream intracellular signaling of integrins shares several components with the activation of the growth factor/growth factor receptor/Ras/Raf/MEK/Erk axis, inducing cell behaviors such as survival, proliferation, and tumor growth [12].

The involvement of integrins with talin can also have repercussions on immune cell regulation. The activation of integrins by cytokines triggers the JAK/STAT pathway, whereby the small family of tyrosine kinase receptors JAK promotes STAT accumulation on the nucleus, which leads to the control of T cell extravasation and, therefore, the regulation of inflammation [13]. In this case, cell sensitivity towards cytokines relied directly upon their ability to bind to integrins and trigger F-actin cytoskeleton reorganization, which in turn can also promote cell polarization, migration, and survival [13].

Although most integrins have one ECM ligand they primarily bind to to generate their effector roles in cell biology, they can also bind to other components with various degrees of affinity, a behavior previously described as promiscuous [14]. For instance, both α5β1 and αvβ3 integrins bind to the RGD motif found in fibronectin, vitronectin, fibrinogen, and collagen I; however, their roles in cell adhesion and motility are different. While αvβ3 integrin is strongly related to the directional persistence of migration, α5β1 integrin is more associated with higher traction forces and strength of adhesion [15,16,17,18]. The inhibition of integrin αvβ3 by a specific protein impairs cell migration and its directionality while also increasing the size and number of adhesion spots [17].

The intimate connection between integrins, the cytoskeleton, and the ECM is linked to cell responses such as migration, a crucial process in many physiological and pathological conditions including embryo development, wound healing, inflammation, and metastasis. The delicate balance between staying attached or moving toward a direction depends on various chemical and mechanical signals from the microenvironment [19]. For instance, neutrophils become adherent to the endothelium and migrate through the ECM toward a site of bacterial infection due to a chemical gradient of inflammatory mediators [20]. For many types of carcinomas, tumor cells acquire a migratory phenotype after undergoing the process of epithelial-mesenchymal transition (EMT), in which epithelial cells lose their apical-basal polarity and change their preset of cell–cell and cell–ECM interactions to a migratory phenotype [21,22]. The migrating tumor cells may enter the blood and lymphatics capillaries and small vessels, where they navigate through the body to search for a new place to adhere to and colonize [23]. This alternating cycle of cell adhesion and propelling migration is an essential part of inflammation and metastasis, as well as other steps of tumor progression in several types of cancer [24,25].

Multiple studies have correlated the upregulation of certain integrins to disease progression and poor prognosis, yet we are unsure if this behavior is caused by or is a consequence of the evolution of cancer. Nevertheless, the overexpression of certain types of integrins in secondary tumors cannot be neglected and it has been considered an attractive target for therapeutic investigation. For instance, overexpression of integrin α5β1 has been correlated with poorer overall survival and prognosis in high-grade serous ovarian cancer [26]. A mechanism of resistance to glioblastoma treatment with temozolomide was also uncovered with a modulation of integrin α5β1 [27]. Among others, integrins β3, β6, α2, and β1 are some of the subunits suggested to identify specific cancers such as melanoma, non-small-cell lung cancer and carcinomas, albeit most research is directed to in vivo, in vitro, or in silico approaches [28,29,30,31].

Integrin αvβ3 and its main ligand vitronectin were correlated with glioblastoma progression and invasiveness in the brain [32]. These findings led to the development and clinical trials of cilengitide, a cyclic peptide inhibitor of αvβ3 and α5β1 integrins. Despite favorable Phase I and II results, the Phase III CENTRIC and Phase II CORE trials did not result in increased patient survival [33]. Nevertheless, further analyses from the CORE study demonstrated a positive association between higher αvβ3 integrin levels and improved survival in patients treated with cilengitide [34]. These data encouraged the continuation of studies on the effects of integrin inhibition on tumor progression. Cosset et al., in 2017, identified a subpopulation of glioblastoma cells dependent on the aberrant activation of the αvβ3 integrin, whose signaling results in increased expression of GLUT3, helping cell survival by facilitating glucose uptake [35]. Inhibition of αvβ3 integrin activation downregulated GLUT3 expression and decreased cell survival and matrix docking independence. The authors concluded that the identification of glioblastoma subtypes could help in defining whether it would be beneficial to treat patients with integrin inhibitors. In addition, other studies suggested that the inefficiency of αvβ3 inhibitors in clinical trials may be due to immune suppression induced by such compounds [36].

## 3. Extracellular Vesicles and Integrins: Perspectives on Drug Delivery and Resistance

One of the main characteristics of cancer cells is their increased biosynthesis of extracellular vesicles (EVs), nano-sized particles released by cells to the extracellular compartment with the potential to transfer biomolecules to different cells and tissues [37]. Their characteristics change according to their biogenesis, composition, origin, and physical-chemical properties, which is indicated in the literature by a vast nomenclature including terms such as exosomes, microvesicles, oncosomes, apoptotic bodies, and others [38]. To optimize EV-related research, a minimal information guide has been curated by experts in the field to suggest the use of a broad nomenclature (i.e., EVs) for any extracellular particle, except when extensive data indicate a specific subtype [39]. In this review, we will maintain the nomenclature employed by the authors, even if they do not reflect MISEV2018 guidelines.

Accumulating evidence suggests that integrins found in EVs may promote a pro-invasive tumor microenvironment, aiding cells in migration [40,41], immune system remodeling [42], and adhesion and uptake [43]. Exosomal integrins may direct organotropic metastasis after uptake by distant cells [44,45], or even create a migratory leading edge following chemotaxis [40]. Integrins αvβ6 and αvβ3 in EVs from prostate cancer were demonstrated to increase cell migration of non-tumorigenic recipient cells [46,47,48]. EVs derived from taxane-resistant prostate cancer cells (PC-3 cells) showed elevated levels of β4 integrins and vinculin which could be useful as biomarkers of disease progression [49]. Targeting integrin α6 by EV-loaded miR-127-3p was shown to inhibit the proliferation and invasion of choriocarcinoma cells in vitro, suggesting a new approach to the treatment of this highly aggressive neoplasia [50].

EVs have also been proposed as biomarkers for specific types of cancer or targets for drug delivery [45,51,52]. A proteomic study compared the EV integrin profiles of a panel of 60 cancer cell lines with non-malignant cell lines, and the results indicated a higher level of integrins in malignant cells, particularly integrins α6, αv, and β1 [45,53]. Moreover, the authors observed a correlation between the high levels of these integrins and disease progression.

More recently, the topic of EV-mediated drug resistance has gained notoriety. Concerning EVs from other tumor-derived cells, such as CAFs, annexin A6 was shown to stabilize integrin β1 via activation of FAK-YAP, which in turn increased drug resistance in gastric cancer cells [54]. Therapy resistance was also observed in a xenograft model subjected to colon and prostate cancer cell-derived EVs, where resistance to anti-PD-L1 was observed [55]. EVs produced by cancer-associated fibroblasts (CAFs) were demonstrated to produce stemness, epithelial-mesenchymal transition, metastasis, and drug resistance when transferred to colorectal cancer [56,57], bladder cancer [58], gastric cancer [54], and pancreatic cancer cells [59]. Similarly, tumor-associated macrophages produce EVs whose cargo directly influences resistance to treatments. Zheng and colleagues (2017) demonstrated that cisplatin resistance in gastric cancer was promoted by macrophage-derived EVs [60]. On the other hand, cisplatin resistance can be transferred by EVs from tumor-resistant to naive recipient cells, as demonstrated for TNBC [61]. Most studies showed that the information in EV cargo is mainly due to microRNAs transferred to recipient cells; however, integrin receptors also have relevant roles in this process, but they are much less studied.

It is noteworthy that studies involving vesicular integrin αvβ3 and drug resistance are scarce. Although it was suggested that exosomal integrins could sequester therapeutics, therefore reducing their blood concentration and reducing efficacy [7], recent advancements show that the blockage of this integrin has a disruptive behavior in EV biogenesis in breast cancer [43].

## 4. Introducing Integrin αvβ3

From the multiple integrin heterodimers, integrin αvβ3 is the RGD-binding vitronectin receptor that may also bind to other ECM ligands including collagen I. It is usually poorly expressed in most cells but can be upregulated in tumor cells and endothelial cells within the tumor microenvironment [62,63]. Our group has demonstrated that the αvβ3 integrin is a key determinant of migration directionality in oral squamous carcinoma cells [17]. Upon αvβ3 integrin blockage, cancer cells moved in circles collectively, indicating a loss of directionality and a change in the migration mode toward a collective and less aggressive cell migration phenotype [17]. Interestingly, fibroblasts apparently do not respond to αvβ3 integrin blockers, despite the expression of significant amounts of the β3 subunit [17,64].

The role of αvβ3 integrin in tumor progression has been extensively studied and demonstrated in several types of cancers, including breast, lung, cervical, and pancreatic [32,65,66]. Ross and colleagues reported that β3 integrin is strongly expressed in breast tumor cell bone metastasis in experimental models but not in the primary tumors or visceral metastases [62]. Similar results were also observed in human patient samples. Due to the integrin expression pattern found in bone metastasis, the authors proposed that αvβ3 integrin targeting can enhance drug delivery to tumor cells within the bone. Indeed, αvβ3 integrin-targeted nanoparticles carrying docetaxel reduced bone metastases and tumor-associated bone destruction more effectively than free docetaxel [62].

Following up on the promiscuity of integrin binding, integrin αvβ3 was also demonstrated as a plasma membrane receptor for estrogen, androgen, and thyroid hormones [67]. Physiological levels of triiodothyronine (T3) and L-thyroxine (T4) bound to integrin αvβ3 promote proliferative and angiogenic behaviors, as the result of the activation of the extracellular signal-regulated kinase 1/2 (ERK1/2), and increased expression of programmed death-ligand 1 (PD-L1) expression [67]. The authors also showed that nanoparticle-bound tetraiodothyroacetic acid (tetrac), an antagonist of T4, can neutralize this epigenetic signaling and inhibit angiogenesis [67]. In a different study, the interaction between integrin αvβ3 and PD-L1 was suggested to regulate the evasion of immune system regulation through the PD-1/PD-L1 axis, upon interferon (IFN) α/β and IFN-γ receptor control [68].

The blockage of integrin αvβ3 has been investigated by different groups as an approach for impairing tumor progression, drug resistance, and metastasis. The use of RGD-binding antagonists of integrin αvβ3 is the most frequent concept and one of the most successful in vitro and in vivo. Bone metastasis of triple-negative breast cancer (TNBC) was inhibited in murine models using peptide PSK1404 [4], an RGD-binding molecule with an affinity toward integrin αvβ3. Chen et al., 2022, demonstrated that doxycycline-induced FAK phosphorylation is blocked by antagonists of the αvβ3 integrin [69].

A recent report indicates that the integrin αvβ3 can also be found in different cellular compartments other than the plasma membrane, which broadens the perspective for integrin αvβ3-based treatments. In particular, a research group from Israel identified that a reservoir of integrin αvβ3 is found in the nuclear membrane of highly metastatic ovarian cancer, with a distinct function from the surface-stationed receptors [70]. Integrins can often be internalized and recycled by cells using multiple pathways yet reports on the presence of nuclear integrin trafficking are still limited [71]. Figure 2 summarizes some of the main roles of αvβ3 integrin in tumor progression and resistance to treatments, which will be discussed below.

## 5. Role of Integrin αvβ3 in Tyrosine Kinase Inhibitor (TKI) Resistance

During the last decade, studies related to the chemoresistance of tumors have intensified, mainly caused by the prolonged survival of patients due to positive advances in therapies. The evolution of cancer treatment has enabled the emergence of targeted therapies, particularly those that focus on tyrosine kinases, nuclear receptors, and other molecular targets, with high success rates. Despite the positive aspects, chemoresistance is a persistent issue that has forced physicians to seek alternatives to prevent disease progression due to drug resistance [1].

Receptor tyrosine kinases (RTKs) are some of the most successful targets for cancer treatment and comprise a family with 58 surface membrane receptors that elicit intracellular responses from extracellular stimuli, through the phosphorylation of tyrosine residues on substrate proteins [72]. Since RTKs are inherently responsible for proliferation and survival, dysfunctions in this system are a staple of many different types of neoplasia. Targeted RTK inhibition by tyrosine kinase inhibitors (TKis), small molecules that inhibit the enzymatic activity of the receptors, alongside monoclonal antibody therapy and other RTK ectodomain blockage approaches, comprise the current state of the art of RTK-targeted therapy [72]. TRKs are often found close to integrins in a crosstalk since neither integrin subunit contains a catalytic domain; integrin αvβ3, for instance, has been described in association with the insulin-like growth factor receptor (IGFR-1), platelet-derived growth factor receptor (PDGFR), and vascular endothelial growth factor receptor (VEGFR2) [73].

Our group has previously demonstrated that the inhibition of integrin αvβ3 generates repercussions in different RTKs. In a metastatic murine TNBC cell line (4T1BM2), integrin αvβ3 blockage by a disintegrin activates the autophagic program through phosphoinositide 3-kinases (PI3K). In this mechanism, cells remain attached to a substrate while the inhibited receptor is recycled, and then resume their initial programming of survival, proliferation, or migration [64]. A different study demonstrated that resistance to neratinib-induced ferroptosis is strongly related to the upregulation of αvβ3 integrin [74]. Ferroptosis is a type of cell death that occurs when there is an imbalance between iron metabolism and the cellular antioxidant system [75], and may be induced by inhibitors of glutathione peroxidase 4 (GPX4) [76], by errors in glutathione cytoplasmic import [77], or by an increased labile iron/Fe^2+^ pool due to impairments on the iron importer transferrin receptor 1 (TFR-1), ferritin, and iron exporter ferroportin-1 [74,78]. Therefore, the inhibition of integrin β3 induced persistent AKT signaling in resistant cells and was involved in iron and antioxidant metabolic reprogramming [74].

Evidence of the role of β3 integrity in drug resistance has been reported in murine bone metastasis of breast cancer after docetaxel chemotherapy [5]. Triple-negative breast cancer patients that underwent neoadjuvant therapy with docetaxel also presented high levels of β3 integrin expression in their bone metastases, which was associated with a worse prognosis [5]. The chemotherapy-induced metabolic response mediated by β3 integrity was suggested to target mTORC1 by the authors and has yet to be confirmed [5]. A different group reported how αvβ3 integrin responds to OSCC resistance to 5-fluorouracil. In this work, OSCC biopsy samples obtained from 30 sensitive and 19 resistant cases revealed osteoporotic (OPN) as the only significantly up-regulated gene, yet a specific integrin inhibitor could reverse this correlation. Therefore, OPN overexpression in OSCC cells resulted in resistance to 5-fluorouracil in an αvβ3 integrin-dependent mechanism [79].

A different study also confirmed the key role of OPN in chemoresistance to platinum treatment in ovarian cancer. The mechanism involves the activation of the ABC drug efflux transporter, and its elevated levels were also clinically associated with poor prognosis [80]. Pharmacological inhibition of OPN improved the efficacy of cisplatin in human and mouse ovarian tumor xenografts, suggesting that OPN might be an attractive target for enhancing platinum sensitivity in ovarian cancer [80]. In acute myeloid leukemia, the interaction between integrin αvβ3 and OPN has also decreased sensitivity to sorafenib [81].

Integrin αvβ3 was also demonstrated as a marker for breast, lung, and pancreatic carcinomas highly resistant to RTK inhibitors (TKI) such as erlotinib in animal models and human patient samples [66]. Chemoresistance involved the activation of the KRAS-RalB-NF-κB axis, triggered by the interaction of the integrin, galectin-3, and KRAS with subsequent downstream activation of NF-κB for the development of a stemness phenotype. Interestingly, the pharmacological inhibition of this pathway could reverse drug resistance [66]. Similarly, the reversion of TKI resistance in lung cancer was also achieved through the overexpression of miR-483-3p [82]. This microRNA, which targets the αvβ3 integrin and can be silenced by hypermethylation of its own promoter, has reverted EMT and inhibited migration, invasion, and metastasis of gefitinib-resistant lung cancer cells [82].

## 6. Immune Checkpoint Inhibitors (ICIs) and the Integrin αvβ3

The class of immune checkpoint inhibitors (ICIs) represents the most recent line of treatment for different types of cancer and has exhibited an improvement in patient care. Infiltrated lymphocytes in the tumor, mainly cytotoxic T cells, correlate with a good prognosis and better response to neoadjuvant therapy. The binding of PD-L1 expressing cells to T cells containing the receptor for PD-L1 (PD-1) is a negative regulator of cytotoxic T cell activity and is therefore considered an important mechanism of evasion of control of tumor growth by the immune system. Hence, the inhibition of the PDL-1/PD-1 axis facilitates lymphocytic cytotoxic activity and enhances immune response [83,84]. This strategy led to the development of ICIs such as atezolizumab, pembrolizumab, and nivolumab. In 2019, atezolizumab was approved by the Food and Drug Administration (FDA, USA) in combination with nab-paclitaxel for the treatment of locally advanced or metastatic TNBC, based on the results of the Mpassion130 phase III clinical trial [85]. Positive data from the KEYNOTE 355 phase III clinical trial led to the approval to use pembrolizumab as a neoadjuvant in cancer treatment [86].

Despite the advancements in the field, ICIs frequently elicit adverse effects in patients because of their lack of specificity to tumor-associated lymphocytes [87]. Therefore, targeting other tumor-enriched receptors such as the integrin αvβ3 is a promising idea to diminish patient discomfort. A study has described that the αvβ3 integrin is a critical regulator of PD-L1 expression and a key component of the tumor immune evasion machinery [68]. The depletion of αvβ3 integrin impaired tumor growth, increased sensitivity to immunotherapy, and allowed for prolonged immunotherapeutic protection [68]. Furthermore, activation of integrin αvβ3 by T4 can also increase the expression of PD-L1 in tumor cells, which in turn aids immune surveillance escape and proliferation. Physiological concentrations of T4 induce the proliferation of several types of tumor cells, including myeloma, glioma, lung carcinoma, and ovarian cancer [88]. On the other hand, TKI therapy may induce the overexpression of thyroid hormones in cancer patients, and this transient hyperthyroidism is associated with poor prognosis [89]. Despite these promising results, the use of integrin αvβ3 as a target for ICIs is poorly investigated and requires further advances to assess the feasibility of this approach.

## 7. The αvβ3 Integrin in Resistance to Radiotherapy

The use of radiotherapy for the treatment of localized solid tumors has been widespread in clinics, with major improvements over the last 25 years due to new technologies and combined therapeutic approaches [90]. A significant consequence of radiotherapy is that cancer cells often develop random mutations in the DNA-damage-response genes, which can be a double-edged sword for patients: tumors can either become unstable due to the stochastic nature of random mutations or may develop radiotherapy resistance and become more aggressive over time [90]. This delicate balance between radiosensitivity and radio resistance in tumor cells can be explored for alternative therapies, including how integrin αvβ3 may respond to these different scenarios.

In prostate cancer cells, the increased expression of integrin αvβ3 upregulated survivin, a small member of the anti-apoptosis protein family responsible for caspase activation. Survivin downregulation was prevented by integrin αvβ3 expression upon ionizing radiation therapy, therefore modulating the radiosensitivity in vitro [91]. In a different study, high αvβ3 integrin expression in patients with cervical cancer that underwent curative radiotherapy was correlated with a poor prognosis [65]. However, in non-small-cell-lung cancer models, the blockage of αvβ3 by its antagonist cilengitide increased the efficacy of radiotherapy [92].

In ovarian cancer cells, the EMT phenotype was induced by integrin αvβ3 activation under T4 stimuli, and ionizing radiation therapy elicited similar results [93]. Regarding cell invasion, Haeger and collaborators (2020) demonstrated via intravital microscopy that collective invasion of orthotopic sarcoma and melanoma in xenograft models have a population of radiotherapy-resistant cells in the leading edge. They also observed that individual inhibitors for integrins β1 and αv led to radio resistance and metastasis development, while dual integrin targeting produced efficient radiosensitization and prevented metastasis [94].

Other integrins, in addition to αvβ3 integrin, are tightly involved in several steps of the metastatic cascade, including the epithelial-to-mesenchymal transition (EMT) [95]. In TNBC, the interaction of integrin β1 with CD146, a highly prevalent protein involved in DNA damage response, was demonstrated to activate the LATS1-YAP pathway and induce radiation resistance [96]. The authors state that this protein is also produced by breast cancer cells after exposure to cisplatin. In other carcinomas, epithelial integrins α6β4 and α6β7 can also mediate additional crosstalk with other cellular receptors, and co-targeting is required to minimize radiation therapy resistance [97].

## 8. Concluding Remarks

Integrins have an important role in cancer progression and resistance to treatment. Due to their generalist nature and data from clinical trials, targeting a single type of integrin is often disadvantageous to patients. Different perspectives on cancer treatment, such as the targeting of integrins in extracellular vesicles, may offer new opportunities for integrin-derived therapies. Consistent results point towards a key role for integrin αvβ3 in the resistance to different cancer therapies, yet the compensatory expression of other similar integrins and adhesion receptors may jeopardize the advancement of treatment for a single receptor. The crosstalk of integrins with other growth factor tyrosine kinase receptors may be used as an approach for alternative therapies, but it may also be a hindrance due to the adaptive response of tumor cells to an aggressive microenvironment. A dual or perhaps multi-targeted integrin approach may be more suitable for avoiding tumor progression and radio, immune, and/or chemotherapy resistance.

## Figures and Tables

**Figure 1 biomedicines-12-01163-f001:**
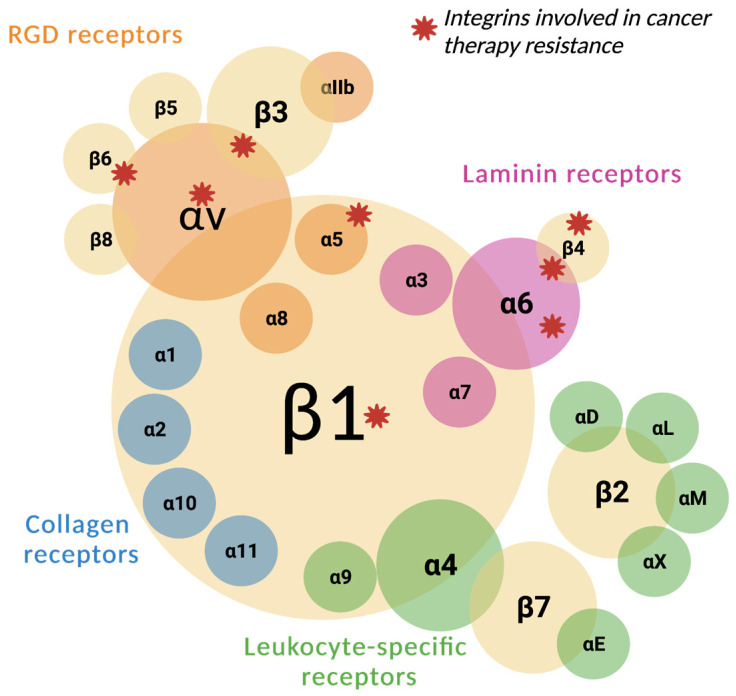
Integrin subunits and their known heterodimers are classified as receptors for RGD (orange), laminin (pink), collagen (blue), and leukocyte-specific receptors (green). Integrins involved in cancer therapy resistance are highlighted in red.

**Figure 2 biomedicines-12-01163-f002:**
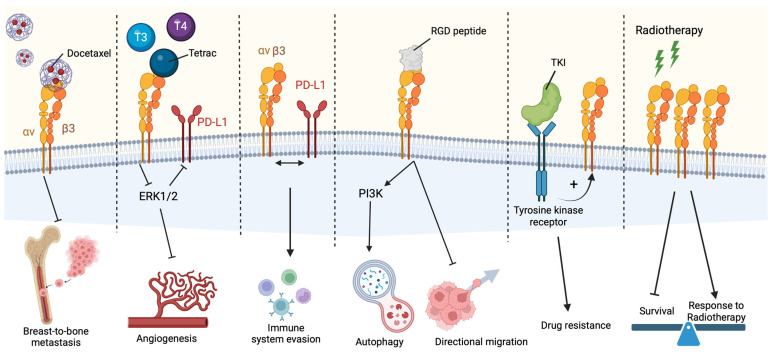
Integrin αvβ3 downstream mechanisms in cancer development and progression based on different stimuli. In summary, docetaxel chemotherapy has an inhibitory effect in αvβ3-mediated breast-to-bone metastasis; tetrac binding to integrins in T3/T4 physiological-level environments inhibits ERK1/2-mediated PD-L1 activation, therefore disrupting angiogenesis; integrin αvβ3 and PD-L1 cross-talk have a role in immune system evasion in cancer; the binding of integrins with RDG peptides activates PI3K-derived autophagy and inhibits directional migration; inhibition of tyrosine kinase receptors increases integrin availability, which promotes drug resistance; radiotherapy stimuli of integrins may act as a double-edged sword, either reducing cell survival or promoting adaptive response to the treatment.

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
