# Peer review of "The Role of αvβ3 Integrin in Cancer Therapy Resistance"

_biomedicines, 2024, doi:10.3390/biomedicines12061163_

Round 1

Reviewer 1 Report

Comments and Suggestions for Authors

This review manuscript describes the role of integrin αvβ3 in cancer resistance. It is concise and interesting. The provided content and figures are accurate. However, the manuscript has some technical shortcomings, which need to be corrected in a revised version:

Both figures of the manuscript are labelled as ´´Figure 1´´. Please correct.

Line 194: Please correct ´´tetraidothyroacetic acid´´.

Line 213: Please write ´´avb3´´ in Greek letters.

Some sentences about the correlation of integrin αvβ3 with JAK-STAT signaling should be added.

Acknowledgments: Was Figure 2 created with the mentioned program, too?

References: The number of references does not match with the numbers shown in the main text. The references section ends with No 62, while in the main text the highest number is 90 (line 351). The authors must solve this problem.

Author Response

Reviewer #1

This review manuscript describes the role of integrin αvβ3 in cancer resistance. It is concise and interesting. The provided content and figures are accurate. However, the manuscript has some technical shortcomings, which need to be corrected in a revised version:

Answer: We thank the reviewer for his critical analysis and for pointing out the flaws in the manuscript. We apologize for that. Changes in the text are highlighted in yellow.

Both figures of the manuscript are labelled as ´´Figure 1´´. Please correct.

Answer: Fixed

Line 194: Please correct ´´tetraidothyroacetic acid´´.

Answer: Fixed

Line 213: Please write ´´avb3´´ in Greek letters.

Answer: Fixed.

Some sentences about the correlation of integrin αvβ3 with JAK-STAT signaling should be added.

Answer: We included this information on page 2 with Reference #13 (text highlighted in yellow).

Acknowledgments: Was Figure 2 created with the mentioned program, too?

Answer: Yes, we used BioRender for creation of the two figures.

References: The number of references does not match with the numbers shown in the main text. The references section ends with No 62, while in the main text the highest number is 90 (line 351). The authors must solve this problem.

Answer: The whole list was double-checked and corrected. Thank you

Reviewer 2 Report

Comments and Suggestions for Authors

In this review, authors summarized the role of αvβ3 integrin in tumor resistance. Overall, the manuscript is easy for readers to understand, and the figures are excellent. The following is my comment.

There are also many reports regarding role of other integrins in cancer such as PMID: 36717550, PMID: 33999338, PMID: 35277328, PMID: 35597804, PMID: 37046674, PMID: 35626034, PMID: 35053532. Authors should add brief descriptions about this point with adding these citations.  

There are reports with immune-related adverse events (irAEs) induced by anti-PD-1 therapy and integrin therapy like PMID: 33517201. With adding this and other citations, authors should add the possible irAEs by targeting therapy to integrin when it is combined with checkpoint inhibitors.

Author Response

Reviewer # 2

In this review, authors summarized the role of αvβ3 integrin in tumor resistance. Overall, the manuscript is easy for readers to understand, and the figures are excellent. The following is my comment.

Answer: We thank the reviewer for his/her critical analysis.

There are also many reports regarding role of other integrins in cancer such as PMID: 36717550, PMID: 33999338, PMID: 35277328, PMID: 35597804, PMID: 37046674, PMID: 35626034, PMID: 35053532. Authors should add brief descriptions about this point with adding these citations.  

Answer: We are aware of the extensive literature on the integrin roles in tumor progression, and therefore, we wanted to keep our focus on the avb3 integrin, which has been our research subject for many years. Even so, we understand that it is important to highlight that other classes of integrins are equally relevant. Therefore, we appreciate the indication of important references, which most of them were included in the text (references 26 to 31).

There are reports with immune-related adverse events (irAEs) induced by anti-PD-1 therapy and integrin therapy like PMID: 33517201. With adding this and other citations, authors should add the possible irAEs by targeting therapy to integrin when it is combined with checkpoint inhibitors.

Answer: The theme of irAEs associated with anti-PDL1 is relatively recent and we could not find data related to avb3 integrin. Most papers refer to case studies with a4b7 inhibitors, including the article cited above. Although the possibility of treating irAEs with anti-integrins is very interesting, we believe it is beyond the scope of this review.

Round 2

Reviewer 1 Report

Comments and Suggestions for Authors

The revised manuscript is suitable for publication now.

Reviewer 2 Report

Comments and Suggestions for Authors

I have no more comments.